# High Performance Attapulgite/Polypyrrole Nanocomposite Reinforced Polystyrene (PS) Foam Based on Supercritical CO_2_ Foaming

**DOI:** 10.3390/polym11060985

**Published:** 2019-06-04

**Authors:** Yidong Liu, Lingfeng Jian, Tianhua Xiao, Rongtao Liu, Shun Yi, Shiyang Zhang, Lingzhi Wang, Ruibin Wang, Yonggang Min

**Affiliations:** 1School of Materials and Energy, Center of Emerging Material and Technology, Guangdong University of Technology, Guangzhou 510006, Guangdong, China; ydliu@gdut.edu.cn (Y.L.); lif23216@163.com (L.J.); xth1011@126.com (T.X.); liu44101@163.com (R.L.); 13290065303@163.com (S.Y.); syz131139@163.com (S.Z.); lingzhiwanglz@126.com (L.W.); 2Dongguan South China Design Innovation Institute, Dongguan 523808, Guangdong, China

**Keywords:** supercritical CO_2_, polystyrene foam, blowing agent, in situ polymerization, attapulgite/polypyrrole nanocomposite

## Abstract

CO_2_ has been regarded as one of the most promising blowing agents for polystyrene (PS) foam due to its non-flammability, low price, nontoxicity, and eco-friendliness. However, the low solubility and fast diffusivity of CO_2_ in PS hinder its potential applications. In this study, an attapulgite (ATP)/polypyrrole (PPy) nanocomposite was developed using the in situ polymerization method to generate the hierarchical cell texture for the PS foam based on the supercritical CO_2_ foaming. The results demonstrated that the nanocomposite could act as an efficient CO_2_ capturer enabling the random release of it during the foaming process. In contrast to the pure PS foam, the ATP/PPy nanocomposite reinforced PS foam is endowed with high cell density (up to 1.9 × 10^6^) and similar thermal conductivity as the neat PS foam, as well as high compression modulus. Therefore, the in situ polymerized ATP/PPy nanocomposite makes supercritical CO_2_ foaming desired candidate to replace the widely used fluorocarbons and chlorofluorocarbons as PS blowing agents.

## 1. Introduction

Polystyrene (PS) foam is one of the most popular and low-cost polymeric foams that is widely used in many applications such as household materials, food containers, lightweight composites, toy models, and packaging [1,2,3,4,5,6]. As is well known, the blowing agent applied during the foaming process is the key factor for achieving PS foam of good quality. Recently, with the rising environmental concerns from currently used blowing agents such as fluorocarbons (FCs) and chlorofluorocarbons (CFCs), carbon dioxide (CO_2_) has attracted tremendous interests from the scientific and industrial communities. Compared with FCs and CFCs, CO_2_ at its supercritical state (*T* = 31 °C and *P* = 73.8 bar/1074 psi) can be competitive because it is inexpensive, non-flammable, nontoxic, environmentally friendly, and chemically inert [7,8,9,10]. However, CO_2_ also has some drawbacks, in that it is usually processed at higher pressures and easily escapes from the polymer matrix, which induces processing instability and shaping contraction. These will result in an uncontrollable foam density and cell morphology of PS foam, thus leading to poor thermal and mechanical properties [11,12,13]. In this case, it is requisite to develop an efficient strategy to manage CO_2_ for PS foaming.

In a traditional process based on supercritical CO_2_ foaming, a large quantity of CO_2_ is dissolved in PS that can modify the rheological properties of PS in the barrel of the extruder, then resulting in extensive expansion during the relaxation at the die. Consequently, the reduction of viscosity decreases the mechanical constraints and the operating temperature within the extruder [14,15,16]. It was found that CO_2_ can decrease the viscosity of PS without otherwise changing their pseudoplastic behavior. The viscosity data for the PS based on supercritical CO_2_ foaming follow the ideal viscoelastic scaling, whereby the set of viscosity curve can be scaled to a master curve of reduced viscosity vs. reduced shear rate identical to the viscosity curve for the pure PS [17]. It was also reported that CO_2_ could blow the PS-based composites. Pang et al. studied the effect of CO_2_ as the processing medium to improve the in situ compatibilization of polypropylene/PS blends via reactive extrusion. CO_2_ had been proved to enable an improved phase dispersion and thus the mechanical properties of the polypropylene/PS through two functions: CO_2_-assisted phase dispersion and CO_2_-promoted in situ compatibilization [18,19]. In addition, the PS/carbon-based nanoparticle composites with different shapes or dimensions were prepared using supercritical CO_2_ as the foaming agent, of which the nucleation mechanism was well analyzed by the classical nucleation theory [20].

In this study, attapulgite (ATP) and polypyrrole (PPy) have been in situ polymerized to give a nanocomposite to assist CO_2_ for PS foaming, where the ATP/PPy nanocomposite is expected to act as a “CO_2_ capturer” to control the CO_2_ releasing during foaming [21]. The ATP-PS foams possessed cells of the similar three-petal shape as the neat PS foam, while the ATP/PPy nanocomposites reinforced PS foams displayed significantly different cellular structure. As demonstrated in the SEM images, it was found that the ATP/PPy nanocomposites reinforced PS foams were composed of multi-petal (≥3) and hierarchical petal-in-petal textures. The foam density and cell size features of the ATP/PPy nanocomposites reinforced PS foams were significantly different from the pure PS foam. This could be understood from the fact that the uniquely fibrous ATP with PPy could form the separated pathways for CO_2_ during the foaming process, interfering with its spontaneous release. As a result, the ATP/PPy nanocomposites reinforced PS foams were endowed with similar thermal conductivity, as well as up to 8.06 °C higher thermal decomposition temperature (TDT) and 181% higher compression modulus, compared to 415.78 °C and 1.09 MPa of the neat PS foam. The ATP/PPy-PS foam with these excellent performances opens enormous opportunities for them to be used in the lightweight composite, microwave absorption, energy, and catalytic applications.

## 2. Materials and Methods

### 2.1. Materials

Pyrrole, Ammonium persulfate (APS), Sodium hexametaphosphate (SHMP), (3-Aminopropyl) triethoxysilane (KH-550), and other chemicals were purchased from Aladdin Chemistry Co., Ltd. (Shanghai, China). All chemicals were of analytical reagent grade used without further purification. PS (Nova 1600) was provided by NOVA Chemical, Inc. (Painesville, OH, USA). Deionized (DI) water was used throughout this study.

ATP was purified from the crude mineral ores (Gansu, China) as follows: First, the crude mineral ores were put into the planetary ball mill (QM3SP4L, Shanghai Xinnuo Instrument Equipment Co., LTD, Shanghai, China) and ball milled at 400 rpm for 8 h. The resulting solid was further ground in an agate mortar for 10 min, followed by sifting fine ATP (400 mesh) from the ores. After that, the fine ATP was dispersed in DI water then ultra-sonicated for 30 min to obtain the uniform dispersion. Subsequently, the ATP dispersion was centrifuged at 5000 rpm for 30 min. Finally, further air-drying was conducted at 105 °C overnight to complete the purification (Figure 1).

### 2.2. Foam Extrusion of ATP/PPy Nanocomposite Reinforced PS

At first, the ATP/PPy nanocomposites were prepared according to the chemical oxidation for the preparation of PPy. Specifically, 0.9 g of purified ATP was dispersed in 500 mL of DI water under magnetic stirring overnight at ambient temperature. Pyrrole monomers of desired amount and 500 mL of HCl (2 M) were subsequently introduced into the previous solution, under vigorous magnetic stirring for another 24 h at 5 °C. Then a mixture of APS (13.02 g) and 100 mL of DI water was added in to initiate the polymerization for about 16 h, during which the temperature of which was kept at 5 °C. The obtained suspension was washed with deionized water three times and centrifuged to obtain the sediment. Finally, the sediment was collected and freeze-dried to give the powdery ATP/PPy nanocomposite (Figure 1c).

The ATP/PPy nanocomposite reinforced PS foams were prepared by a modified method as originally presented by Castro and co-workers [10]. Foaming was done using a twin screw extruder equipped with a pelletizer die as shown in Figure 2 (Leistritz ZSE-27; *D* = 27 mm; *L*/*D* = 40). The temperature zones of the extruder barrel were kept at 160 °C in the feed zone and 180 °C at the die tip. The extruder was typically run at 100 rpm. The 2.0, 2.8, and 3.3 wt% of ATP/PPy nanocomposite were selected to reinforce the PS foams, thus being labeled as ATP/PPy-PS 1, 2, and 3, respectively. For a comparison purpose, pristine PS and PS reinforced by the purified ATP alone were also prepared, denoting as PS foam and ATP-PS 1, 2, 3 (with respectively 2.0, 2.8, and 3.3 wt% of purified ATP included). The visual observations of neat PS, ATP-PS 3, and ATP/PPy-PS are presented in Figure 3 (optical camera images).

The cell size and cell density of all PS foams (Figure 4, scanning electron microscopy, SEM images) were determined by using both the inverted microscope (equipped with a 4× objective lens, ECLIPSE Ti2-U, Nikon Corporation, Tokyo, Japan) and SEM (S3400N, Hitachi Co., Ltd., Tokyo, Japan). At least 70 cells were taken into account to determine the average cell size and cell density. The obtained morphological properties and densities of the neat PS foam, ATP-PS foams, and ATP/PPy-PS foams are summarized in Table 1. Specifically, the foam density of each sample was calculated according to the following equation from the ASTM D792 standard [22]:(1)$$FoamDensity=\frac{M_{air}\times \rho_{water}}{M_{air}-M_{water}}$$ where *M*_air_ is the apparent mass of the sample in air, *M*_water_ is the apparent mass of the sample in water, and *ρ*_water_ is the density of water.

### 2.3. Thermal and Mechanical Properties

Thermogravimetric analysis (TGA) was carried out using a Simultaneous TGA/differential scanning calorimetry (DSC) analyzer (TGA/DSC 3+, Mettler Toledo, Greifensee, Switzerland) with temperature scanning from ambient to 800 °C at a heating ramp of 10 °C/min. Besides this, TDT for all samples was defined as the temperature where 50% of weight loss was viewed. Also, the thermal stability of samples was evaluated by differential scanning calorimetry (DSC, TGA/DSC 3+, Mettler Toledo, Switzerland) while flowing N_2_ over a range of 25–800 °C and with a heating rate of 10 °C/min. In addition, the thermal conductivity was evaluated by using Thermal Constants Analyzer (TPS-500s, Hot Disk AB Co., Gothenburg, Sweden) at room temperature under 20 mW within 10 s.

The compressive properties measurements were carried out using an electronic universal testing machine (Inspekt table blue 5 kN, Hegewald & Peschke, Nossen, Germany) at a testing speed of 1 mm/min for all samples according to the ASTM D1621-16 standard. The sample size is 40 mm × 20 mm × 16 mm.

All the above tests were performed on at least five samples from each set of measurements.

## 3. Results and Discussion

### 3.1. Foam Morphology

The morphology of the foam samples was examined to investigate the effect of the ATP/PPy nanocomposites on CO_2_ for foaming PS. The optical images compare the cell size and cell density of PS foam with others, which illustrate the ATP-PS foam is made up of many three-petal flowers, which is similar to that viewed for neat PS foam, whereas the ATP/PPy-PS foam exhibits significantly different multi-petal and hierarchical-petal flower textures (Figure 4a–c). The unique texture of ATP/PPy-PS foam was further determined by SEM (Figure 4d–f). With the addition of ATP/PPy nanocomposite, the obtained foams displaying increased cell density than both the PS foam and the ATP-PS foam as graphed in Figure 4g. In contrast to 3.5 × 10^3^ of the PS foam, the cell density of the ATP/PPy-PS foams increased 500-fold to within 3.5 × 10^5^–1.9 × 10^6^. Meanwhile, as listed in Table 1, both the foam density and average cell size for the ATP/PPy-PS foams were decreased to 0.052–0.058 g/cm^3^ and 0.19–0.27 mm, respectively, from 0.132–0.188 g/cm^3^ and 0.40–0.56 mm for the ATP-PS foam. To our knowledge, this may be attributed to that the positive effect brought by overdosage of ATP or ATP/PPy can cause many drawbacks such as low diffusion and long contact time on the supercritical CO_2_ foaming [23].

On the basis of these results, we propose that the reinforcement of ATP/PPy nanocomposites may be attributed to the physical confinement of them. On one hand, after the inclusion of the nanocomposites, the initial geometry of PS during foaming was changed as the nanocomposites would occasionally stand in the way where the PS cells growing. As a result, it is possible for CO_2_ to get split or disorder, correspondingly generating the cells of multi-petal and hierarchical-petal flower textures. This constraint effect of the nanocomposites can be understood by regarding them as the CO_2_ capturer to manage its releasing other than the usual manner [24,25,26]. On the other hand, both the rod-like ATP and fibrous PPy may mislead CO_2_ to flow on more directions. Herein, ATP functions as a good substrate for other polymers, such as PPy in this study or polyaniline in our previous papers, to get improved performances [27,28].

### 3.2. Thermal and Mechanical Properties

The thermal properties are crucial for PS-based materials. As addressed above, the incorporation of ATP or ATP/PPy nanocomposites in PS can facilitate the random releasing of CO_2_, however, the increased thermal conductivity was simultaneously observed (Figure 5a). The highest thermal conductivity was of 0.054 W/(mK) for ATP-PS 1, 38% higher than 0.039 W/(mK) of the neat PS foam, which hinders the potential application of ATP-PS, though further increasing the content of ATP in the ATP-PS foams exhibits a decreased trend. This may be due to that, although ATP is a thermally conductive clay (~0.68 W/(mK)), it is also easy to get agglomerated to embody decreased heat transfer. In other words, the obtained ATP-PS foams with higher ATP content sacrificed their thermal conductivity to gain increased applicable potential. Note that the thermal conductivity curve of the ATP/PPy-PS foams is much different from that of the ATP-PS foams, which linearly grows along with the content of the ATP/PPy nanocomposite. On one hand, the inclusion of comparably thermally non-conductive PPy in the foam can compromise ATP. On the other hand, the growth of fibrous PPy on ATP in the nanocomposites avoids ATP to agglomerate [29,30]. Besides this, TGA is applied to estimate the thermal stability of these foams according to their thermogravimetric behavior (Figure 5b). As shown in Figure 5c, it can be clearly observed that all foams shared similar TGA curve patterns, while ATP/PPy-PS 1 exhibited the highest TDT, which is ~8 °C higher than that of the PS foam. Encouragingly, all ATP/PPy-PS foams outperformed the ATP-PS foams with respect to TDT. This manifests that ATP has a limited effect on PS here, unless nanocomposited with PPy to form the thermostable interactions between Fe_2_O_3_ (7.53 wt%) in ATP and PPy chains [31]. Increasing the dosage of ATP shows a similar trend as seen in Figure 5a, suggesting that the inclusion of highly thermally conductive ATP in the foams facilitates their decomposition under heating. Note that the TDT curve of the ATP/PPy-PS foams is similar to that of ATP-PS foams, though significantly different from its thermal conductivity data. These effects most likely due to the relatively thermal stable dopant anion of PPy, which can increase the TDT of the ATP/PPy-PS foams [32]. In addition, the thermal behavior of all foams was also evaluated by DSC. As elucidated in Figure 5d, the DSC profiles of the PS foam, the ATP-PS foams, and the ATP/PPy-PS foams are alike, indicating that they were based on the same degradation mechanism as the decompositions of neat PS [33].

As a foam material, mechanical properties are also highly desired. Figure 6a shows the compression modulus of all foams; the ATP-PS foams and the ATP/PPy-PS foams present up to 4.18 (383%) and 3.06 MPa (281%), respectively, much higher than 1.09 MPa (100%) of the neat PS foam. The compression modulus data of the ATP-PS foams and the ATP/PPy-PS foams are inconsistent with the cell density value of them as listed in Table 1. This may be due to that the calculated cell density calculation concerns all cells in three dimensions, whereas the compression modulus is only directly related to the vertical dimension. Furthermore, the open-cell percentage of the ATP/PPy-PS foams varies a lot from the others, 63.71–78.96% vs. 33.95% and 31.00–40.07% (Figure 6b). This difference can be clearly told in the respective top view SEM images as demonstrated in Figure 6c–i. As reported earlier, materials with open cellular structures are known to exhibit low modulus [34]. Therefore, we believe that when the close-cell statue is maintained, the compression modulus of the ATP-PS foams can be significantly increased from that of the neat PS foam, in agreement with the changing trend of their cell density. The introduction of aromatic tertiary amine groups on the surface of ATP leads to the parallel alignment of aromatic rings in the PS chains that conduces to the effective load transfer between the ATP surface and PS matrix [35]. On the other hand, once it is reversed into the open-cell statue, as happened on the ATP/PPy-PS foams, the contribution of the significantly changed cell density to the compression modulus may be compromised to some extent. Nonetheless, the highly open-cell ATP/PPy-PS foams still outperform their nonporous or poorly porous counterparts in regard to the application of microwave absorption as well as the accessibility of the active surface of the materials [36,37,38,39,40].

## 4. Conclusions

In summary, the in situ polymerized attapulgite/polypyrrole nanocomposite has been successfully introduced into the supercritical CO_2_-based PS foam. The optical graphs and SEM images of all foams demonstrated that even though the typical foam structure of PS was formed throughout the others, of which the foam density and cell size features were significantly different. By combining the nanocomposites, the thus-obtained foams exhibited the cells of significantly different multi-petal and hierarchical-petal flower textures, along with more than 500 times higher (1.9 × 10^6^) cell density than the pure PS foam. This may be attributed to that the nanocomposite could act as an efficient CO_2_ capturer so enables the random release of it during the foaming process. Also, the thermal behavior and mechanical properties of the ATP/PPy-PS foams were investigated, which were endowed with up to 8.06 °C higher TDT and 281% higher compression modulus than those of the neat PS foam. Overall, the ATP/PPy nanocomposite reinforced PS foams fabricated in this study will open up numerous opportunities for a range of applications based on the supercritical CO_2_ foaming.

## Figures and Tables

**Figure 1 polymers-11-00985-f001:**
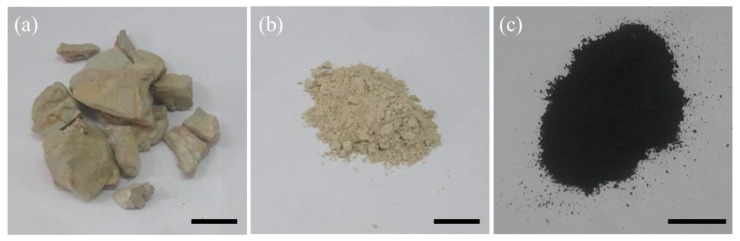
Digital camera images of (**a**) the crude mineral ore of attapulgite (ATP), (**b**) purified ATP and (**c**) the ATP/polypyrrole (PPy) nanocomposite. Scale bar: 1 cm.

**Figure 2 polymers-11-00985-f002:**
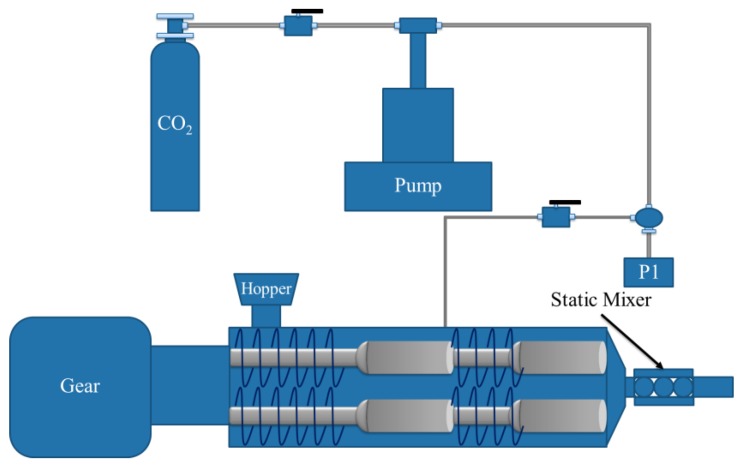
The continuous foaming extrusion process for the fabrication of the polystyrene (PS) foam in this study.

**Figure 3 polymers-11-00985-f003:**
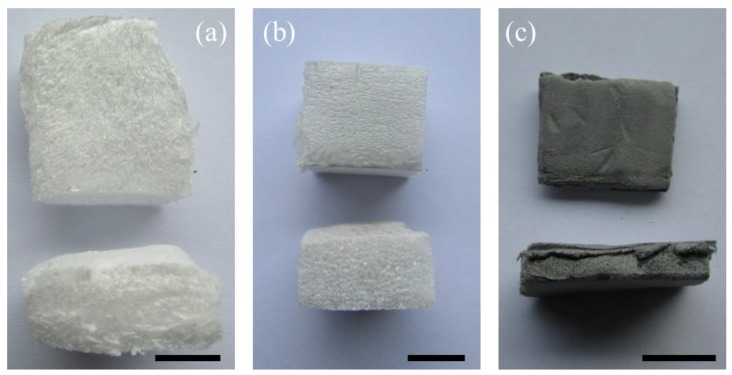
Optical camera images (upper: top view, lower: cross section view) of (**a**) neat PS foam, (**b**) ATP-PS 3, (**c**) ATP/PPy-PS 3. Scale bar: 1 cm.

**Figure 4 polymers-11-00985-f004:**
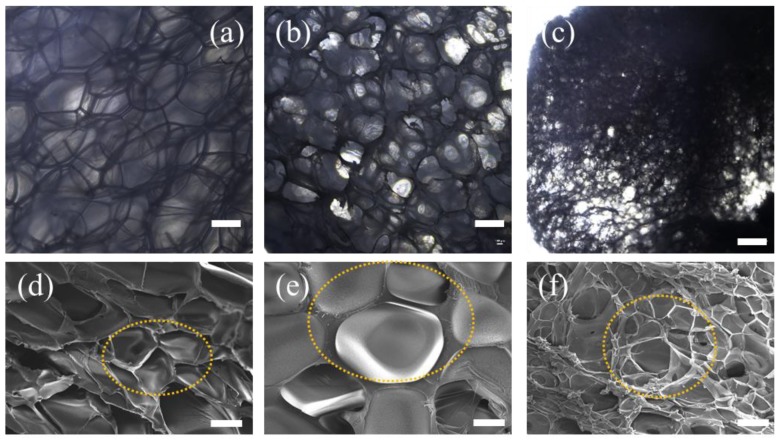
The optical microscopy images showing the cell size and cell density of (**a**) neat PS foam, (**b**) ATP-PS 3 and (**c**) ATP/PPy-PS 3, scale bar: 500 μm. Top view scanning electron microscopy (SEM) micrographs of (**d**) neat PS foam, (**e**) ATP-PS 3 and (**f**) ATP/PPy-PS 3, scale bar: 2 μm. (**g**) Cell density test of ATP-PS foam and ATP/PPy-PS foams.

**Figure 5 polymers-11-00985-f005:**
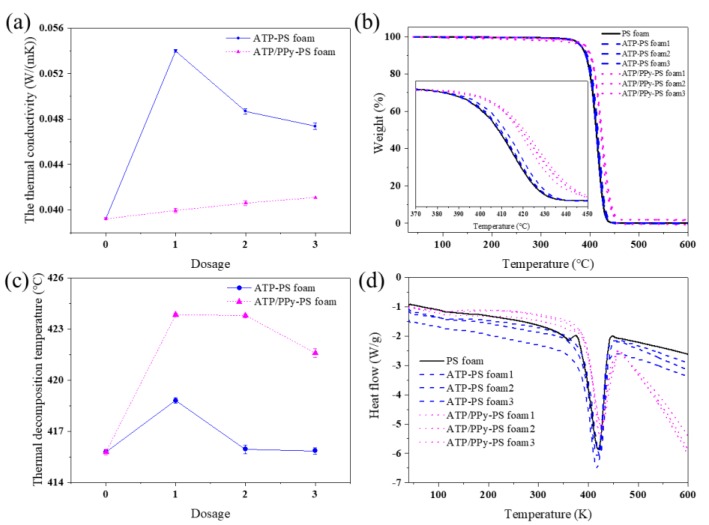
(**a**) Thermal conductivity, (**b**) thermogravimetric analysis (TGA, the partial curves within the temperature range of 370–450 °C were zoomed in as shown in the inset), (**c**) thermal decomposition temperature (TDT) and (**d**) differential scanning calorimetry (DSC) analyses of neat PS foam, ATP-PS foams, and ATP/PPy-PS foams, respectively.

**Figure 6 polymers-11-00985-f006:**
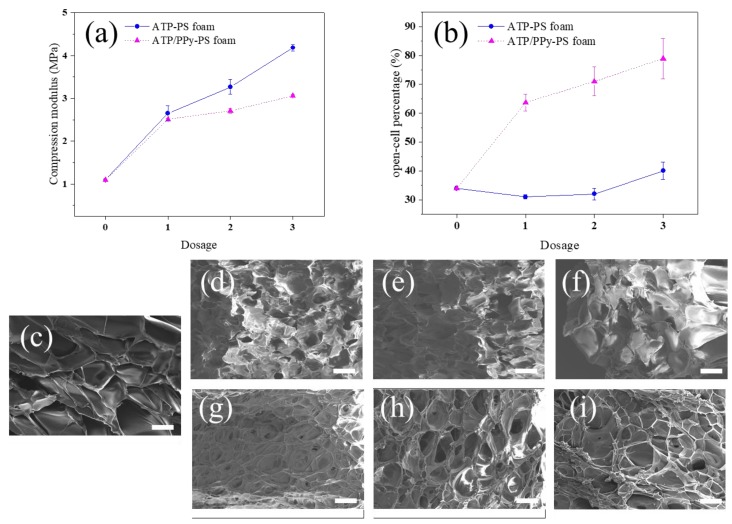
(**a**) The compression modulus measurements of neat PS foam, ATP-PS foams, and ATP/PPy-PS foams, according to the ASTM D1621-16 standard. (**b**) Corresponding open-cell percentage of all foams, with respective top view SEM images of (**c**) neat PS foam, (**d**–**f**) ATP-PS 1, 2, 3 and (**g**–**i**) ATP/PPy-PS 1, 2, 3, scale bar: 2 μm.

**Table 1 polymers-11-00985-t001:** Density and morphological properties of neat PS foam, ATP-PS foam, and ATP/PPy-PS foam.

Sample	Foam Density (g/cm^3^)	Cell Size (mm)	Cell Density (cells/cm^3^)
neat PS	0.070 ± 0.004	0.52 ± 0.06	(3.5 ± 0.2) × 10^3^
ATP-PS 1	0.188 ± 0.015	0.40 ± 0.03	(8.3 ± 0.2) × 10^4^
ATP-PS 2	0.132 ± 0.008	0.53 ± 0.02	(4.2 ± 0.2) × 10^4^
ATP-PS 3	0.159 ± 0.016	0.56 ± 0.04	(1.3 ± 0.1) × 10^4^
ATP/PPy-PS 1	0.052 ± 0.004	0.19 ± 0.01	(1.9 ± 0.1) × 10^6^
ATP/PPy-PS 2	0.058 ± 0.007	0.27 ± 0.02	(3.7 ± 0.2) × 10^5^
ATP/PPy-PS 3	0.055 ± 0.003	0.22 ± 0.01	(3.5 ± 0.1) × 10^5^

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
