# Peer review of "High Performance Attapulgite/Polypyrrole Nanocomposite Reinforced Polystyrene (PS) Foam Based on Supercritical CO2 Foaming"

_polymers, 2019, doi:10.3390/polym11060985_

Round 1
Reviewer 1 Report
This manuscript does not meet the expected requirements of a scientific publication due to major issues and errors. The abstract is informative and gives a good overview to the current work. Please emphasize the relevance of this work, as the set benchmarks are far worse that the actual “state of the art”-material as well as the not existing necessity to add expensive fillers to a mainly low-priced material (Foam extrusion of PS works pretty well with CO2). Foaming PS with CO2 is hence not a sufficient motivation and basically not the problem. Long-time storage of small blowing agent molecules in PS is the main problem, that can be addressed.
Additionally, the authors, as well as any scientist working in the field of foams, should know that polystyrene is not a good candidate to be used in the automotive sector, because of the extreme flammability of PS (and depending on the process harmful blowing agents). Also, only open cell foams are used for sound insulation (mainly PUR). Further misunderstandings arise, as the authors discuss batch foaming (row 58) even though extrusion foaming was carried out.
Also, the main advantage of PS-foams is having the superior insulation properties due to their low thermal conductivity. Hence, emphasize an increase in thermal conductivity as a positive effect is misleading and confusing. Also, the use of the word “synthesized” implies an actual own synthesis, which in this case is only compounding. The authors carried out tensile measurements according to experimental section with foams and reported the values of compression modulus. This is an obvious mistake, as it is either the young’s modulus or they should conduct compression tests to determine proper compression modulus.Contradicting values for neat PS foam densities in the text and in the table, too small micrographs (unclear and unrecognizable) and comparing foams with different densities (density is the primary influence on mechanics) are worsening the insufficient quality of this written work. Comparing similar density ranges of neat PS and ATP/PPy-PS-foams show nonsignificant increase in Young’s-Modulus
The quality of the language is not satisfactory with a lot of grammatical mistakes and not understandable passages (row 66-69). The abstract is informative though and gives a good overview to the current work. The results have to be extended and confirmed by thorough measurements and proper reference materials to make the research more systematic, scientific and realistic for industrial / actual applications.
I would recommend that this article is not sufficient enough to be published on Polymers at the current state. The suggested improvements should be carried out, additional proper testing with an emphasized relevance for actual application should be established and finally reviewed again.
Additional Annotation to the Review
Page 3, Row 3, Are PE foams that widely used? Has to be rechecked and proven with the source of information. PE foams rather bad, and PP foams application ist expected to be way higher
Page 3, Row 9, Make => makes
Page 3, Row 13, Deteriorates seriously => seriously sound not scientific
Page 5, Row1, Ultra-high=> your expansion is similar to established industry product, as you claimed => therefore misleading
Page 9, Row 9-11, Example for an already known fact
Page 9, Row 16-17, Example for sentence with 3 x is
Page 10, Figure s, Line to guide the eyes
Page 13, Row 7-8, High melt strength at low temperature? How low, still melt? Source of this information? Cells are difficult to grow up=> cell growth is inhibited / restricted
Page 14, Figure s, Line to guide the eyes
Page 14, Row 2-3, Sentence is not comprehensive
Page 16, Row 9, Co2 is difficult to diffuse => Diffusion into the sample is inhibited/ aggravated
Page 17, Chapter 3.4 is really difficult to read
Page 20, Figure s, Strain in %, and refer the curves to the density, not only expansion ratio
Author Response
Response to Reviewer 1 Comments
Point 1:
Please emphasize the relevance of this work, as the set benchmarks are far worse that the actual “state of the art”-material as well as the not existing necessity to add expensive fillers to a mainly low-priced material (Foam extrusion of PS works pretty well with CO2). Foaming PS with CO2 is hence not a sufficient motivation and basically not the problem. Long-time storage of small blowing agent molecules in PS is the main problem, that can be addressed.
Response 1:
This comment is very good. As we discussed in the main text, the inclusion of ATP/PPy nanocomposite could significantly improve the supercritical CO2 foaming of PS, simultaneously bringing much better mechanical property than pristine PS. In this case, even though foam extrusion of PS works pretty well with CO2, further adding ATP/PPy nanocomposite in can obtain the PS foam with better performance. Moreover, the fillers here are not so expensive. The highest addition of ATP/PPy nanocomposite for the foams was 3.3 wt%, included in which there is only 0.33% of PPy. Based on the newly found ATP mineral pit with the proved reserves up to 1,000,000,000 tons in Gansu Province of China, the price of ATP will sharply decrease soon. For the long-time storage of small blowing agent molecules in PS, which is actually addressed in this manuscript. In detail, as shown in Figure 4d-e, the ATP-PS foam is endowed with relatively smoother surface than neat PS foam, which may be attributed to the needle-like ATP can affect the diffusion of the CO2 molecules during the foaming process. In Figure 4f, the ATP/PPy-PS foam displays a much higher cell density, suggesting that the ATP/PPy nanocomposite changes the way and extends the storage time of CO2 molecules during the foaming process.
Point 2:
Additionally, the authors, as well as any scientist working in the field of foams, should know that polystyrene is not a good candidate to be used in the automotive sector, because of the extreme flammability of PS (and depending on the process harmful blowing agents). Also, only open cell foams are used for sound insulation (mainly PUR). Further misunderstandings arise, as the authors discuss batch foaming (row 58) even though extrusion foaming was carried out.
Response 2:
Sorry for these mistakes. In line 31 of Page 1, “sound insulation” and “auto parts” have been placed by “food container” and “toy models”. In line 56-60 on Page 2, “Except for using alone, co-blowing agents such as n-pentane, cyclopentane, and water can join CO2 in batch foaming to improve the performances of PS foam” has been replaced by “In addition, the PS/carbon-based nanoparticle composites with different shapes or dimensions were prepared using supercritical CO2 as the foaming agent, of which the nucleation mechanism was well analyzed by the classical nucleation theory [20].”.
Point 3:
Also, the main advantage of PS-foams is having the superior insulation properties due to their low thermal conductivity. Hence, emphasize an increase in thermal conductivity as a positive effect is misleading and confusing.
Response 3:
The first paragraph of section 3.2 on Page 6 has been revised as follows: The thermal properties are crucial for PS-based materials. As addressed above, the incorporation of ATP or ATP/PPy nanocomposites in PS can facilitate the controlled releasing of CO2, however, the increased thermal conductivity was simultaneously observed (Figure 5a). The highest thermal conductivity was of 0.054 W/(mK) for the ATP-PS 1, 38 % higher than 0.039 W/(mK) of the neat PS foam, which hinders the potential application of ATP-PS, though further increasing the content of ATP in the ATP-PS foams exhibits a decreased trend. This negative effect of dosage may be due to that although ATP is a thermally conductive clay (~0.68 W/(mK)), meanwhile it is easy to get agglomerated under increased addition thus results in weaker heat transfer. This can also be confirmed as partially replacing ATP by PPy, the thus-obtained ATP/PPy-PS foams only give a slightly increasing thermally conductive curve, in comparison to that of the ATP-PS foams. On one hand, the inclusion of comparably thermally non-conductive PPy in the foam can compromise ATP. On the other hand, the growth of fibrous PPy on ATP in the nanocomposites avoids ATP to agglomerate [28,29]. Besides, TGA is applied to estimate the thermal stability of these foams according to their thermogravimetric behavior (Figure 5b). As shown in Figure 5c, it can be clearly observed that all foams shared similar TGA curve patterns, while ATP/PPy-PS1 exhibits the highest TDT that is ~8 °C higher than that of the PS foam. Encouragingly, all ATP/PPy-PS foams outperform the ATP-PS foams with respect to TDT. This manifests that ATP has limited effect on PS here, unless nanocomposited with PPy to form the thermostable interactions between Fe2O3 (7.53 wt%) in ATP and PPy chains [30]. Increasing the dosage of ATP shows a similar trend as seen in Figure 5a, suggesting that the inclusion of highly thermally conductive ATP in the foams facilitates their decomposition under heating. Note, the TDT curve of the ATP/PPy-PS foams is similar to that of ATP-PS foams, though significantly different from its thermal conductivity data. These effects most likely due to the relatively thermal stable dopant anion of PPy, which can increase the TDT of the ATP/PPy-PS foams [31]. In addition, the thermal behavior of all foams were also evaluated by DSC. As elucidated in Figure 5d, the DSC profiles of the PS foam, the ATP-PS foams and the ATP/PPy-PS foams are alike, indicating that they were based on the same degradation mechanism as the decompositions of neat PS [32].
Point 4:
Also, the use of the word “synthesized” implies an actual own synthesis, which in this case is only compounding.
Response 4:
In line 94 and 104 on Page 3, “Synthesis of ATP/PPy nanocomposite” and “were synthesized by” have been revised as “Compounding of ATP/PPy nanocomposite” and “were prepared by”, respectively.
Point 5:
The authors carried out tensile measurements according to experimental section with foams and reported the values of compression modulus. This is an obvious mistake, as it is either the young’s modulus or they should conduct compression tests to determine proper compression modulus.
Response 5:
This is a typo. In line 139-142 on Page 4, the whole paragraph has been revised as “The compressive properties measurements were carried out using an electronic universal testing machine (Inspekt table blue 5 kN, Hegewald & Peschke, Germany) at a testing speed of 1 mm/min for all samples according to the ASTM D1621 - 16 standard. The sample size is 40 mm × 20 mm × 16 mm”.
Point 6:
Contradicting values for neat PS foam densities in the text and in the table, too small micrographs (unclear and unrecognizable) and comparing foams with different densities (density is the primary influence on mechanics) are worsening the insufficient quality of this written work.
Response 6:
(i) The contradicting values have been revised as follows: In line 23 on Page 1, 3.7 ×105 has been corrected as 1.9 ×106; in line 154 on Page 5, “100-fold to within 1.9×105-3.7×105” has been corrected as “500-fold to within 3.5×105-1.9×106”; in the third and fifth lines of Table 1 on Page 5, “4.2±0.2 ×105” and “1.9±0.1 ×105” have been corrected as “1.9 ×106” and “1.9 ×106”. (ii) In line 172 on Page 5, Figure 4 has been re-organized to give clearer observation. (iii) For “the comparison of foams with different densities” question, it was our thoughtlessness that failed to explain well. In this case, more characterizations had been conducted to address our points. In line 214-233 on Page 7, the whole paragraph includes Figure 6 has been revised and re-organized as “As a foam material, mechanical properties are also highly desired. Figure 6a shows the compression modulus of all foams, the ATP-PS foams and the ATP/PPy-PS foams present up to 4.18 (383 %) and 3.06 MPa (281 %), respectively, much higher than 1.09 MPa (100%) of the neat PS foam. The compression modulus data of the ATP-PS foams and the ATP/PPy-PS foams are not in consistent with the cell density value of them as listed in Table 1. This may be due to that the calculated cell density calculation concerns all cells in three dimensions, whereas the compression modulus is only directly to the vertical dimension. Furthermore, the open-cell percentage of the ATP/PPy-PS foams varies a lot from the others, 63.71-78.96 % vs 33.95 % and 31.00-40.07 % (Figure 6b). This difference can be clearly told in the respective top view SEM images as demonstrated in Figure 6c-i. As reported earlier, materials with open cellular structures are known to exhibit low modulus [33]. Therefore, we believe that when the close-cell statue is maintained, the compression modulus of the ATP-PS foams can be significantly increased from that of the neat PS foam, in agreement with the change trend of their cell density. The introduction of aromatic tertiary amine groups on the surface of ATP leads to the parallel alignment of aromatic rings in the PS chains that conduces to the effective load transfer between the ATP surface and PS matrix [34]. On the other hand, once it is reversed into the open-cell statue, as happened on the ATP/PPy-PS foams, the contribution of the significantly changed cell density to the compression modulus may be compromised to some extents. Nonetheless, the highly open-cell ATP/PPy-PS foams still outperform their nonporous or poorly porous counterparts in regards to the application of microwave adsorbing as well as accessibility of the active surface of the materials [35]. Figure 6. (a) The compression modulus measurements of neat PS foam, ATP-PS foams and ATP/PPy-PS foams, according to the ASTM D1621-16 standard. (b) Corresponding open-cell percentage of all foams, with respective top view SEM images of (c) neat PS foam, (d-f) ATP-PS 1, 2, 3 foam and (g-i) ATP/PPy-PS 1, 2, 3 foam, scale bar: 2 μm”.
Point 7:
Comparing similar density ranges of neat PS and ATP/PPy-PS-foams show nonsignificant increase in Young’s-Modulus.
Response 7:
As responded to point 5, the compression properties of all samples had been redone according to the ASTM D1621-16 standard. The obtained compression modulus for neat PS, ATP-PS-foams and ATP/PPy-PS form 1, 2, 3 were 1.09 ± 0.01, 2.65 ± 0.17 & 3.27 ± 0.17 & 4.18 ± 0.07, and 2.51 ± 0.02 & 2.71 ± 0.06 & 3.06 ± 0.03, MPa. There were 143, 199, 283, 130, 148, 180 % of enhancement of compression modulus viewed for the ATP-PS-foams and ATP/PPy-PS forms, respectively, comparing with the neat PS. As already reported, the introduction of aromatic tertiary amine groups on the surface of ATP leads to the parallel alignment of aromatic rings in the PS chains that conduces to the effective load transfer between the ATP surface and PS matrix (Journal of Applied Polymer Science, 2015, 132(9)). However, PPy in the ATP/PPy-PS foams increased the percentage of open-cell feature to up to 70 %, in contrast to 32 % for the neat PS and 33 % for the ATP-PS foams (Figure 4d-f), leading to the cellular structure with weaker compression modulus.
Point 8:
The quality of the language is not satisfactory with a lot of grammatical mistakes and not understandable passages (row 66-69). The abstract is informative though and gives a good overview to the current work.
Response 8:
Line 66-69 on Page 2 has been revised as “The ATP-PS foams possessed cells of the similar three-petal shape as the neat PS foam, while the ATP/PPy nanocomposites reinforced PS foams displayed significantly different cellular structure. As demonstrated in the SEM images, it was found that the ATP/PPy nanocomposites reinforced PS foams were composed of multi-petal (≥ 3) and hierarchical petal-in-petal textures”.
Point 9:
The results have to be extended and confirmed by thorough measurements and proper reference materials to make the research more systematic, scientific and realistic for industrial / actual applications.
Response 9:
This issue has been carefully addressed in response 5-7.
Point 10:
Additional Annotation to the Review Page Row Issue 3 3 Are PE foams that widely used? Has to be rechecked and proven with the source of information. PE foams rather bad, and PP foams application is expected to be way higher 9 Make => makes 13 Deteriorates seriously => seriously sound not scientific 5 1 Ultra-high=> your expansion is similar to established industry product, as you claimed => therefore misleading 9 9-11 Example for an already known fact 16-17 Example for sentence with 3 x is 10 Figures Line to guide the eyes 13 7-8 High melt strength at low temperature? How low, still melt? Source of this information? Cells are difficult to grow up=> cell growth is inhibited /restricted 14 Figures Line to guide the eyes 2-3 Sentence is not comprehensive 16 9 CO2 is difficult to diffuse => Diffusion into the sample is inhibited/ aggravated 17 Chapter 3.4 is really difficult to read 20 Figures Strain in %, and refer the curves to the density, not only expansion ratio
Response 10:
Thanks for the careful work of the reviewer. May I sincerely ask the reviewer to recheck this point, many of which are not present in this manuscript.

Reviewer 2 Report
The author introduces reinforced PS foam, ways of preparing the foam using nanocomposite, their morphologies and properties in this article. This article shows the next generation of PS foam having high thermal and mechanical properties by controlling CO2 release during the foaming process. Unfortunately, this review has absence of some information and some elements that interfere with reader’s deep understanding. Therefore, we believe that the current version of this review article is lacking in the great accomplishment of these research fields. We provide major revision of this article for publishing in ‘Polymers’ journal. Some questions and suggestions are as follows;
[1] In Fig. 4g, the ATP-PS foam and ATP/PPy-PS foam tend to increase the cell density in dosage 1, and tend to decrease in dosage 2 and 3. Therefore, we suggest authors add the discussion about this tendency to increase understanding this manuscript by readers.
[2] Authors showed the thermal conductivity and the thermal decomposition temperature in Fig. 5a and Fig. 5c, respectively. The thermal conductivity and thermal decomposition temperature of ATP-PS foam and ATP/PPy-PS foam has different trend according to the foam type such as formulation. We suggest that authors to explain why this data shows a different trend.
[3] The author did not describe how to measure each foam density in Table 1. We ask you to describe measurement method of foam density.
[4] The authors showed three digital camera images of ATP in Fig. 1. However, the description of each picture is lacking in this manuscript. We suggest that a caption for (c) should be provided.
[5] We could not find any scale bar in Figs. 1 and 3. Therefore, we suggest that the scale bar needs to be added into Figs. 1 and 3 for readers.
[6] It is recommended to capitalize the first alphabetical word in Title except preposition.
[7] We suggest that authors should label the ATP-PS and ATP/PPy-PS in Table 1.
[8] The captions in Fig. 5b and Fig. 5c are reversed. We suggest that you modify the caption correctly.
[9] The caption for Fig. 6a and Fig. 6b are incorrect. We suggest that you modify the caption correctly.
[10] If the abbreviation is used for the first time in manuscript, we suggest that authors name it with the full-name. : Line No.73 TDT
[11] In this manuscript, there are some parts that do not conform to typographical or grammatical. We suggest authors to modify correctly. Errors and corrections are listed below.
: Line No.57 in situ – in-situ
: Line No.153 PS – ATP-PS
: Fig. 4 (f) caption ATP-PS – ATP/PPy-PS
: Line No.198 … ATP-PS 2 – ATP-PS 1
: Line No.210 Nonethelss – Nonetheless
[12] The form of the reference does not match with guideline of this journal. The authors should modify references based on guideline. It is recommended that authors modify the reference form by referring to the URL address below. https://www.mdpi.com/authors/references
Author Response
Response to Reviewer 2 Comments
Point 1:
In Fig. 4g, the ATP-PS foam and ATP/PPy-PS foam tend to increase the cell density in dosage 1, and tend to decrease in dosage 2 and 3. Therefore, we suggest authors add the discussion about this tendency to increase understanding this manuscript by readers.
Response 1:
“In addition, further increasing the ATP or ATP/PPy addition in PS decreases the cell density of all obtained foams. To our knowledge, this may be attributed to that the positive effect brought by overdosage of ATP or ATP/PPy can cause many drawbacks such as low diffusion and long contact time on the supercritical CO2 foaming” has been added in the line 159 on Page 5, to discuss the trend.
Point 2:
Authors showed the thermal conductivity and the thermal decomposition temperature in Fig. 5a and Fig. 5c, respectively. The thermal conductivity and thermal decomposition temperature of ATP-PS foam and ATP/PPy-PS foam has different trend according to the foam type such as formulation. We suggest that authors to explain why this data shows a different trend.
Response 2:
The first paragraph of section 3.2 on Page 6 has been revised as follows: The thermal properties are crucial for PS-based materials. As addressed above, the incorporation of ATP or ATP/PPy nanocomposites in PS can facilitate the controlled releasing of CO2, however, the increased thermal conductivity was simultaneously observed (Figure 5a). The highest thermal conductivity was of 0.054 W/(mK) for the ATP-PS 1, 38 % higher than 0.039 W/(mK) of the neat PS foam, which hinders the potential application of ATP-PS, though further increasing the content of ATP in the ATP-PS foams exhibits a decreased trend. This negative effect of dosage may be due to that although ATP is a thermally conductive clay (~0.68 W/(mK)), meanwhile it is easy to get agglomerated under increased addition thus results in weaker heat transfer. This can also be confirmed as partially replacing ATP by PPy, the thus-obtained ATP/PPy-PS foams only give a slightly increasing thermally conductive curve, in comparison to that of the ATP-PS foams. On one hand, the inclusion of comparably thermally non-conductive PPy in the foam can compromise ATP. On the other hand, the growth of fibrous PPy on ATP in the nanocomposites avoids ATP to agglomerate [28,29]. Besides, TGA is applied to estimate the thermal stability of these foams according to their thermogravimetric behavior (Figure 5b). As shown in Figure 5c, it can be clearly observed that all foams shared similar TGA curve patterns, while ATP/PPy-PS1 exhibits the highest TDT that is ~8 °C higher than that of the PS foam. Encouragingly, all ATP/PPy-PS foams outperform the ATP-PS foams with respect to TDT. This manifests that ATP has limited effect on PS here, unless nanocomposited with PPy to form the thermostable interactions between Fe2O3 (7.53 wt%) in ATP and PPy chains [30]. Increasing the dosage of ATP shows a similar trend as seen in Figure 5a, suggesting that the inclusion of highly thermally conductive ATP in the foams facilitates their decomposition under heating. Note, the TDT curve of the ATP/PPy-PS foams is similar to that of ATP-PS foams, though significantly different from its thermal conductivity data. These effects most likely due to the relatively thermal stable dopant anion of PPy, which can increase the TDT of the ATP/PPy-PS foams [31]. In addition, the thermal behavior of all foams were also evaluated by DSC. As elucidated in Figure 5d, the DSC profiles of the PS foam, the ATP-PS foams and the ATP/PPy-PS foams are alike, indicating that they were based on the same degradation mechanism as the decompositions of neat PS [32].
Point 3:
The author did not describe how to measure each foam density in Table 1. We ask you to describe measurement method of foam density.
Response 3:
This is a good comment. In line 121 on Page 3, “Specifically, the foam density of each sample was calculated according to following equation from the ASTM D792 standard [22]: eqs.1 where Mair is the apparent mass of the sample in air, Mwater is the apparent mass of the sample in water, ρwater is the density of water” has been added.
Point 4:
The authors showed three digital camera images of ATP in Fig. 1. However, the description of each picture is lacking in this manuscript. We suggest that a caption for (c) should be provided.
Response 4:
In line 91 on Page 3, “and (b) purified ATP” has been revised as “(b) purified ATP and (c) the ATP/PPy nanocomposite”; in line 102 on Page 3 “(Figure 1c)” has been added after “ATP/PPy nanocomposite”.
Point 5:
We could not find any scale bar in Figs. 1 and 3. Therefore, we suggest that the scale bar needs to be added into Figs. 1 and 3 for readers.
Response 5:
The scale bar has been added in each figure. Also each caption has been revised by adding “Scale bar: 1 cm.” and “Scale bar: 1 cm.”, respectively, followed the original caption.
Point 6:
It is recommended to capitalize the first alphabetical word in Title except preposition.
Response 6:
This advice is very good, every initial letter of the word in the title has been capitalized.
Point 7:
We suggest that authors should label the ATP-PS and ATP/PPy-PS in Table 1.
Response 7:
For Table 1, the first column has been revised as “Sample neat PS ATP-PS 1 ATP-PS 2 ATP-PS 3 ATP/PPy-PS 1 ATP/PPy-PS 2 ATP/PPy-PS 3”.
Point 8:
The captions in Fig. 5b and Fig. 5c are reversed. We suggest that you modify the caption correctly.
Response 8:
The captions in Fig. 5b and 5c have been exchanged as “TGA (the partial curves within the temperature range of 370-450 °C were zoomed in as shown in the inset)” and “TDT”.
Point 9:
The caption for Fig. 6a and Fig. 6b are incorrect. We suggest that you modify the caption correctly.
Response 9:
The caption in line 213-214 for Fig 6 has been corrected as “The deformation (a) and the compression modulus (b) measurements of neat PS foam, ATP-PS foam and ATP/PPy-PS foam.”.
Point 10:
If the abbreviation is used for the first time in manuscript, we suggest that authors name it with the full-name. : Line No.73 TDT.
Response 10:
In line 73, “8.06 °C higher TDT” has been revised as “8.06 °C higher thermal decomposition temperature (TDT)”, as well as in line 129, “the thermal decomposition temperature (TDT)” has been revised as “TDT”
Point 11:
In this manuscript, there are some parts that do not conform to typographical or grammatical. We suggest authors to modify correctly. Errors and corrections are listed below. : Line No.57 in situ – in-situ : Line No.153 PS – ATP-PS : Fig. 4 (f) caption ATP-PS – ATP/PPy-PS : Line No.198 … ATP-PS 2 – ATP-PS 1 : Line No.210 Nonethelss – Nonetheless
Response 11:
Sorry for these mistakes, all of which have been corrected as advised.
Point 12:
The form of the reference does not match with guideline of this journal. The authors should modify references based on guideline. It is recommended that authors modify the reference form by referring to the URL address below. https://www.mdpi.com/authors/references.
Response 12:
Thank you for this good comment, all references have been revised by referring to the reference style downloaded from the mentioned URL address.

Reviewer 3 Report
I believe this is a very good paper and the research is important. I detected some places where the grammar could be improved. Please check the grammar carefully one more time for minor improvements.
Page 1 line 17 that necessarily 17 require hierarchies remove these two words
Page 1 line 21 so enables change to enabling
Page 1 line 25 an alluring change to desired
Page 1 line 32 ubiquitous change these word
Page 1 line 37 change Compare to compared
Page 3 Figure 1 has three images a, b and c but the legend only mentions a and b
Author Response
Response to Reviewer 3 Comments
Point 1: Page 1 line 17 that necessarily 17 require hierarchies remove these two words.
Response 1: “that necessarily require hierarchies” has been removed as advised.
Point 2: Page 1 line 21 so enables change to enabling.
Response 2: “so enables” has been revised as “enabling”.
Point 3: Page 1 line 25 an alluring change to desired.
Response 3: “an alluring” has been revised as “desired”.
Point 4: Page 1 line 32 ubiquitous change these word.
Response 4: “ubiquitous” has been replaced by “widely-used”.
Point 5: Page 1 line 37 change Compare to compared.
Response 5: “Compare” has been corrected as “Compared”.
Point 6: Page 3 Figure 1 has three images a, b and c but the legend only mentions a
and b.
Response 6: Sorry for this mistake, in line 91 on Page 3, “and (b) purified ATP” has been revised as “(b) purified ATP and (c) the ATP/PPy nanocomposite”; in line 102 on Page 3 “(Figure 1c)” has been added after “ATP/PPy nanocomposite”.

Round 2
Reviewer 1 Report
After additional improvements, this article meets the basic requirements of a scientific publication. Still, a lot of issues remain. The abstract is informative and gives a good overview to the current work, even better in the improved state. Still, the significance of the ATP/PPy is not clear / con not be retraced by the reader, who has a basic understanding of foams. Most of the achieved results can be reproduced with basic nucleating agents as Talcum. Hence, the necessity of ATP/PPy and the advantages are not unique. Additionally, the terminus “molecular reservoir” anticipates higher gas sorption or lower desorption, which are not addressed in this work and thus are misleading. Continuous foam extrusion of PS with CO2 has been state of the art for many years and is absolutely unproblematic with the right setup (a lot of experiences on this topic by reviewer department). Cell density and cell morphology are usually adjusted through nucleating agents. Long-time storage of small blowing agent molecules in PS is the main problem, that can be addressed.
The quality of the language is overall satisfactory with a few mistakes. The abstract is informative though and gives a good overview to the current work.
This article is publishable after major improvement! The suggested improvements should be carried out, additional the focus of this work / of the used ATP / PPy (advantages) should be revised.
Additional Annotation to the Review
Page 1
Row 21 Molecular reservoir => anticipates sorption/desorption results
Row 39 Remove “that” from the sentence
Row 40 Inertness => inert
Row 42-44 Asumption, that foaming PS with supercritical CO2 is challenging => No, state of the art. Easiest foaming process.
Page 2
Row 67 Molecular reservoir => anticipates sorption/desorption results
Row 81 Remove “and”
Row 83 Assumption of microwave absortption => why? Reference or evidence?
Row 84 Assumption of catalytic applications => why? Reference or evidence?
Page 3
Row 103 Foam extrusion of ……reinforced PS.
Row 114 Replace “compounding” with “foaming”
Page 4
Row 125 Refer rather to Figure 4, as it relates to morphology evaluation
Page 5
Row 168 For proper comparison, a benchmark system with basic nucleating agent is required => evaluation of efficiency of ATP/PPy
Row 182 Molecular reservoir => anticipates sorption/desorption results
Row 182 If understood correct, the advantage of so called “controlled” CO2 release is resulting through platelets shape of nanoclay =>
A) Thus, any filler in platelet shape can achieve same results
B) Mechanism of “”CO2 splitting”” not “controlled” => rather random with probably main orientation in extrusion direction due to random dispersion
Page 7
Row 198 “controlled” => not controlled, rather random
Row 203-207 Bad language / phrasing
Row 212-215 Higher TDT is an obvious effect of incorporation of anorganic filler => control test
Page 8
Row 266 Assumption of microwave absortption => why? Reference or evidence?
Page 9
Row 282 Molecular reservoir => anticipates sorption/desorption results
Author Response
Response to Reviewer 1 Comments
Point 1: After additional improvements, this article meets the basic requirements of a scientific publication. Still, a lot of issues remain. The abstract is informative and gives a good overview to the current work, even better in the improved state. Still, the significance of the ATP/PPy is not clear / con not be retraced by the reader, who has a basic understanding of foams. Most of the achieved results can be reproduced with basic nucleating agents as Talcum. Hence, the necessity of ATP/PPy and the advantages are not unique.
Response 1: This is a very good comment. Different from other nucleating agents as Talcum (which is of triclinic crystal system), ATP has a unique octahedral structure (as shown in the below image) that can be used to capture CO2, as well addressed in the reported paper (Materials Letters, 2017, 194: 107-109), based on which the thereafter-obtained ATP- and ATP/PPy-PS foams are endowed with increased cell density and porous texture, as well as higher compression modulus. Besides, as a conductive polymer, further inclusion of PPy into PS can give it considerable applicable potential. In addition, the price of ATP is also an advantage, which is lower than 1/4 of that of Talcum and cheaper than most of other nucleating agents, rendering it a promising candidate for the large-scale production of high performance PS foam with low cost.
The schematic diagram of the octahedral structure of ATP.
Point 2: Additionally, the terminus “molecular reservoir” anticipates higher gas sorption or lower desorption, which are not addressed in this work and thus are misleading.
Response 2: Appreciated this useful question, it should be “CO2 capturer” rather than “molecular reservoir”, which has been well addressed in the reported paper (Materials Letters, 2017, 194: 107-109, which has been cited as ref 21#). In detail, ATP has a unique octahedral structure that can be used as cages to temporarily capture CO2. There were also many oxygenated groups (epoxy, hydroxyl) in ATP, thus the inclusion of which could draw enhanced electrostatic interactions to CO2 molecules, which were further strengthened once conductive PPy was introduced as the ATP/PPy-PS foams embodied higher cell density than that of the ATP-PS foams. In addition, the abundant pores on ATP are beneficial for the diffusion of CO2 molecules.
Point 3: Continuous foam extrusion of PS with CO2 has been state of the art for many years and is absolutely unproblematic with the right setup (a lot of experiences on this topic by reviewer department). Cell density and cell morphology are usually adjusted through nucleating agents. Long-time storage of small blowing agent molecules in PS is the main problem, that can be addressed.
Response 3: Even though the continuous foam extrusion of PS with supercritical CO2 has been developed well, there are still some issues need to be addressed so result in the fabrication of PS with better performance. We agree that long-time storage of small blowing agent molecules in PS is one of the main problems. However, in this work, our focus is on how to tailor the morphology of the cells of PS foam by low-cost ATP and conductive PPy, consequently the obtained high performance PS foam are applicable to widened fields.
Point 4: The quality of the language is overall satisfactory with a few mistakes. The abstract is informative though and gives a good overview to the current work.
This article is publishable after major improvement! The suggested improvements should be carried out, additional the focus of this work / of the used ATP / PPy (advantages) should be revised.
Additional Annotation to the Review
Page 1
Row 21 Molecular reservoir => anticipates sorption/desorption results
Row 39 Remove “that” from the sentence
Row 40 Inertness => inert
Row 42-44 Asumption, that foaming PS with supercritical CO2 is challenging => No, state of the art. Easiest foaming process.
Page 2
Row 67 Molecular reservoir => anticipates sorption/desorption results
Row 81 Remove “and”
Row 83 Assumption of microwave absortption => why? Reference or evidence?
Row 84 Assumption of catalytic applications => why? Reference or evidence?
Page 3
Row 103 Foam extrusion of ……reinforced PS.
Row 114 Replace “compounding” with “foaming”
Page 4
Row 125 Refer rather to Figure 4, as it relates to morphology evaluation
Page 5
Row 168 For proper comparison, a benchmark system with basic nucleating agent is required => evaluation of efficiency of ATP/PPy
Row 182 Molecular reservoir => anticipates sorption/desorption results
Row 182 If understood correct, the advantage of so called “controlled” CO2 release is resulting through platelets shape of nanoclay =>
A) Thus, any filler in platelet shape can achieve same results
B) Mechanism of “”CO2 splitting”” not “controlled” => rather random with probably main orientation in extrusion direction due to random dispersion
Page 7
Row 198 “controlled” => not controlled, rather random
Row 203-207 Bad language / phrasing
Row 212-215 Higher TDT is an obvious effect of incorporation of anorganic filler => control test
Page 8
Row 266 Assumption of microwave absortption => why? Reference or evidence?
Page 9
Row 282 Molecular reservoir => anticipates sorption/desorption results
Response 4: Page 1
Row 21 “a stable molecular reservoir for CO2” has been revised as “an efficient CO2 capturer”;
Row 39 “that” has been removed from the sentence;
Row 40 Inertness has been corrected as inert;
Row 42-44 Even though foaming PS with supercritical CO2 is a state of the art, there are still some issues need to be addressed to result in the fabrication of PS foam with better performance so it can be more widely used;
Page 2
Row 67 Molecular reservoir has been revised as “CO2 capturer”;
Row 81 “and” has been removed;
Row 83 Two reported papers (J. Mater. Chem., 2012, 22, 18772-18774) and (ACS Appl. Mater. Interfaces 2013, 5, 3, 883-891) regarding to the PS foam applicable for microwave absorption have been cited where necessary;
Row 84 Two reported papers (J. Phys. Chem. C 2016, 120, 45, 25935-25944) and (J. Mater. Chem. A, 2016, 4, 10810-10815) regarding to the PS foam applicable for microwave absorption have been cited where necessary;
Page 3
Row 103 The title of section 2.2 has been revised as “Foam extrusion of ATP/PPy nanocomposite reinforced PS” as advised;
Row 114 “compounding” has been replaced by “foaming”;
Page 4
Row 125 “Figure 3” has been revised as “Figure 4”;
Page 5
Row 168 This confusing sentence has been removed;
Row 182 Molecular reservoir has been revised as “CO2 capturer”;
Row 182 Yes, the advantage of so called “controlled” CO2 release is resulting through platelets shape of nanoclay =>
A) Thus, the filler used here is ATP that has a unique octahedral structure that can be used to capture CO2 so the thereafter-obtained ATP- and ATP/PPy-PS foams are endowed with increased cell density and porous texture, as well as higher compression modulus. Besides, as a conductive polymer, further inclusion of PPy into PS can give it considerable applicable potential. The above two advantages have been rarely reported concerning the production of high performance PS foam;
B) “controlled” has been revised as “random”;
Page 7
Row 198 “controlled” has been revised as “random”;
Row 203-207 The bad language / phrasing has been carefully revised as “This may be due to that although ATP is a thermally conductive clay (~0.68 W/(mK)), but it is also easy to get agglomerated to embody decreased heat transfer. In other words, the obtained ATP-PS foams with higher ATP content sacrificed their thermal conductivity to gain increased applicable potential. Note, the thermal conductivity curve the ATP/PPy-PS foams is much different in contrast to that of the ATP-PS foams, which linearly grows along with the content of the ATP/PPy nanocomposite.”;
Row 212-215 This is self-evident as the neat PS foam exhibited lower TDT than those of ATP-PS, even for ATP-PS 1 that with as low addition of ATP as 2.0 wt%. To our knowledge, the neat PS can be regarded as a control test without any inclusion of ATP or ATP/PPy nanocomposite;
Page 8
Row 266 The reported papers (J. Mater. Chem., 2012, 22, 18772-18774) and (ACS Appl. Mater. Interfaces 2013, 5, 3, 883-891) those are regarding to the PS foam applicable for microwave absorption have been cited;
Page 9
Row 282 “a stable molecular reservoir for CO2” has been revised as “an efficient CO2 capturer”.
